# Biomarkers of Tumor Heterogeneity in Glioblastoma Multiforme Cohort of TCGA

**DOI:** 10.3390/cancers15082387

**Published:** 2023-04-20

**Authors:** Garrett Winkelmaier, Brandon Koch, Skylar Bogardus, Alexander D. Borowsky, Bahram Parvin

**Affiliations:** 1Department of Electrical and Biomedical Engineering, College of Engineering, University of Nevada Reno, 1664 N. Virginia St., Reno, NV 89509, USA; gwinkelmaier@unr.edu (G.W.);; 2Department of Biostatics, College of Public Health, Ohio State University, 281 W. Lane Ave., Columbus, OH 43210, USA; 3Department of Pathology, UC Davis Comprehensive Cancer Center, University of California Davis, 1 Shields Ave, Davis, CA 95616, USA; 4Pennington Cancer Institute, Renown Health, Reno, NV 89502, USA

**Keywords:** Glioblastoma Multiforme, tumor heterogeneity, biomarker, whole slide imaging, TCGA

## Abstract

**Simple Summary:**

Identifying biomarkers of survival from a large-scale cohort of Glioblastoma Multiforme (GBM) pathology images is hindered by heterogeneity of tumor signature compounded by age being the single most important confounder in predicting survival in GBM. The main contributions of this manuscript are to define (i) metrics for identifying tumor subtypes of tumor heterogeneity and (ii) relevant statistics for incorporating age for evaluating competing hypotheses. As a result, the GBM cohort are stratified based on interpretable morphometric features with or without preconditioning on published genomic subtypes.

**Abstract:**

Tumor Whole Slide Images (WSI) are often heterogeneous, which hinders the discovery of biomarkers in the presence of confounding clinical factors. In this study, we present a pipeline for identifying biomarkers from the Glioblastoma Multiforme (GBM) cohort of WSIs from TCGA archive. The GBM cohort endures many technical artifacts while the discovery of GBM biomarkers is challenged because “age” is the single most confounding factor for predicting outcomes. The proposed approach relies on interpretable features (e.g., nuclear morphometric indices), effective similarity metrics for heterogeneity analysis, and robust statistics for identifying biomarkers. The pipeline first removes artifacts (e.g., pen marks) and partitions each WSI into patches for nuclear segmentation via an extended U-Net for subsequent quantitative representation. Given the variations in fixation and staining that can artificially modulate hematoxylin optical density (HOD), we extended Navab’s Lab method to normalize images and reduce the impact of batch effects. The heterogeneity of each WSI is then represented either as probability density functions (PDF) per patient or as the composition of a dictionary predicted from the entire cohort of WSIs. For PDF- or dictionary-based methods, morphometric subtypes are constructed based on distances computed from optimal transport and linkage analysis or consensus clustering with Euclidean distances, respectively. For each inferred subtype, Kaplan–Meier and/or the Cox regression model are used to regress the survival time. Since age is the single most important confounder for predicting survival in GBM and there is an observed violation of the proportionality assumption in the Cox model, we use both age and age-squared coupled with the Likelihood ratio test and forest plots for evaluating competing statistics. Next, the PDF- and dictionary-based methods are combined to identify biomarkers that are predictive of survival. The combined model has the advantage of integrating global (e.g., cohort scale) and local (e.g., patient scale) attributes of morphometric heterogeneity, coupled with robust statistics, to reveal stable biomarkers. The results indicate that, after normalization of the GBM cohort, mean HOD, eccentricity, and cellularity are predictive of survival. Finally, we also stratified the GBM cohort as a function of EGFR expression and published genomic subtypes to reveal genomic-dependent morphometric biomarkers.

## 1. Introduction

The tumor signature observed in whole slide imaging (WSI) is often heterogeneous, which reflects a complex gene expression program that is unique to each patient. Tumor heterogeneity (TH) can be based on distinct morphological and phenotypic profiles, such as morphology and gene expression. TH is also a strong factor in the tumor burden, with implications for patients’ prognosis and treatment. The goal of our study is to investigate whether biomarkers of tumor heterogeneity can be captured based on computed nuclear indices and their organization in WSIs of the Glioblastoma Multiforme (GBM) dataset in The Cancer Genome Atlas (TCGA). GBM is a Grade IV cancer with a five-year survival rate of 9% [1], in which TH should play an important role. The genomic subtypes of GBM have been characterized [2], providing additional constraints for heterogeneity analysis. However, characterizing heterogeneity is not without challenges in a TCGA dataset, as there may be artifacts in WSI (e.g., pen marks), technical variations in sample preparation and staining, and computational strategies for associating heterogeneity to the outcome need to be developed. Furthermore, the confounding factor of age is the single most important variable in predicting survival in GBM. Therefore, any prediction of biomarkers must incorporate rigorous statistical criteria for validation. In fact, learning survival from histology images has been quite challenging, and coupling with CNN has continued to make incremental improvements [3].

Analysis of WSI has benefited from the integration of various technologies, including whole slide scanning, annotations, and filtering to remove artifacts. Recent advances in computational histopathology, which is based on cytological analysis (such as nuclear atypia and cellular density) and automated feature learning, have also contributed significantly to this field. These techniques enable classification, such as tumor grading and detection of micrometastasis, or association via regression to an outcome such as survival. Deep learning is now the preferred method for image-based analysis and representation [4]. For instance, cytological analyses use nuclear segmentation that extends U-Net [5,6] coupled with adversarial training [7], which is highly effective, particularly in identifying vesicular nuclear phenotypes [8] that traditional methods [9] could not detect. Although a thorough review of nuclear segmentation and feature-based representation for computational histopathology is beyond the scope of this manuscript, this article provides a summary of several studies focused on the analysis of low-grade glioma and GBM.

Mobadersany et al. [10] developed a pipeline for training a modified VGG19 model to learn features and associate them with survival using manually selected regions of interest from a WSI. Each image is assigned a risk vector for input to a Cox proportional hazards layer, which computes a loss function for model construction [3]. This approach was applied to diffused gliomas, resulting in a significant prognostic outcome. Chen et al. [11] integrated CNN for feature-based representation, graph-convolutional network following nuclear segmentation, and an attention mechanism to predict tumor grading or survival. Their pipeline was also applied to diffused gliomas with improved statistical analysis. Kong et al. [12] stratified patients based on transcriptomics and morphometric indices computed from tumor biopsy and histology, respectively. Zhang et al. [13] used multi-kernel learning to integrate histopathology and multi-OMIC data (e.g., gene expression, methylation) to perform prognostic tasks. However, the performance of these techniques is hindered as a result of (a) the absence of color normalization across the cohort, (b) not adjusting for age as a strong confounder, (c) failing to incorporate TH in predictive models, and (d) not addressing the sole special needs of GBM. Although some researchers have stratified GBM patients based on aggressive versus non-aggressive therapies and/or integration with molecular data [9,12], the strict utility of histology and TCGA clinical data, such as age, has been lacking. Moreover, from a translational perspective, it would be more valuable to predict the outcome from a low-cost histology section or, at most, coupled with one or two transcripts from an interpretative representation (e.g., nuclear chromatin content, cellularity). These are the issues we aim to investigate.

The overall process and an example of TH are shown in Figure 1. Each WSI underwent artifact and background removal, followed by segmentation of nuclei in each patch, and optical density-based normalization for feature extraction. An important step was to normalize the color images, based on an extension from Navab’s Lab, to ensure optical density could be used as a biomarker. Computed indices were then combined to represent TH using two alternative strategies for computing subtypes. These subtypes were then analyzed rigorously using a likelihood ratio and forest plots to predict survival. The premise for using alternative representations is that differences in TH can be quantified, revealing phenotypes that are typically masked by averaging or other higher-order statistics. Section 2 outlines the computational methodologies for representing WSIs, while Section 3 lists statistically significant biomarkers predicted using morphometric analysis, both in the absence and presence of genomic subtypes. Finally, Section 4 provides additional insights into our findings and concludes the manuscript.

## 2. Methods

### 2.1. Preprocessing the WSIs

The GBM cohort contains not only technical variations due to sample preparation and staining, but also technical anomalies such as pen marks and out-of-focus images. Initially, we attempted to filter out pen marks using PyHist [14], which utilizes an edge filter and graph cut segmentation. However, this approach did not effectively identify many regions containing pen marks. Therefore, we chose to annotate a dataset of pen marks and create a support vector machine (SVM) classifier that identifies patches of 224-by-224 containing pen marks based on their concatenated RGB pixel histogram (i.e., a vector of 768 × 1). With 625 annotated images and a 90–10 training and testing split, we achieved a training accuracy of 97%. PyHist was effective in removing other artifacts (e.g., blur) and white regions (e.g., background), and we also used it to partition each WSI into 224-by-224 regions.

### 2.2. Nuclear Segmentation

Segmentation of nuclei in H&E stained images can be challenging due to technical variations such as sample preparation and staining, biological heterogeneity such as nuclear atypia and pleomorphism, and variations in Hematoxylin optical density (HOD) and texture such as vesicular phenotype. This important topic has been addressed by numerous researchers [6,7,8]. In our implementation, we modified an earlier approach for segmenting 3D organoids [15] using the U-Net architecture with a modified loss function that integrates a potential field for delineating touching nuclei. We annotated and extended H&E stained images from previous datasets [8,9] and complemented the data augmentation step with local/global contrast adjustment for nuclei/background optical density. For local contrast adjustment, we randomly selected between 10–20% of nuclei and modulated their color intensities, as well as the background regions. This step was crucial because the GBM cohort is diverse in terms of nuclear chromatin or protein contents, contributing to tumor heterogeneity. We started with 57 annotated training images, each no less than 1000-by-700 pixels. Following data augmentation, the sample size increased to 200,000 patches, each sized 224-by-224. Using the leave-one-out method, we computed an Aggregated Jaccard Index (AJI) [16] of 0.62 for the 57 annotated images.

### 2.3. Image Normalization

The TCGA histology cohort lacks standardization in terms of staining, which may not be a significant issue for preprocessing or segmentation, but can affect feature extraction. To address this, we proposed that an improved color normalization approach would yield a more reliable HOD and protein readout for biomarker discovery. We applied a state-of-the-art technique in color normalization [17] to normalize WSIs across the entire cohort, which involves incorporating an L1 regularization term in the loss function for non-negative matrix factorization (NMF) to estimate the source image’s stain matrix, and then mapping the deconvolved image back into a RGB space using a target image’s stain matrix. We also utilized the nuclear mask to aid in rapid convergence and ensure consistent ordering of the two stained channels following NMF. The color normalization method [17] maps each candidate H&E-stained image in the RGB space to a single target image that corresponds to the desired staining, after which we use NMF to estimate HOD. An example of color normalization is shown in Figure 2. In the results section, we compared this approach with a classical method for color decomposition [18] and found that rigorous normalization associates HOD as a statistically validated biomarker predictive of survival.

### 2.4. Computation of the Morphometric Indices

A number of morphometric indices per nucleus were computed following color correction. Indices included were: nuclear size; HOD content; cellularity (a measure of cellular density computed from Delaunay Triangulation); eccentricity (e.g., elongation) (a measure of spindle geometry); and solidity (a measure of nuclear pleomorphism).

### 2.5. Association of the Morphometric Indices with Survival

Predicting biomarkers, based on heterogeneity, requires a representation and a distance metric for computing stable clusters or tumor subtypes. Each tumor subtype is then examined for whether it is predictive of survival.

#### 2.5.1. Representation

Our motivation is to capture heterogeneity while maintaining reduced dimensionality, for example, using a single morphometric index at a time, in support of interpretation. To achieve this, we chose to represent each index either as a PDF or an ensemble of vocabularies. In the first case, a “cohort PDF” was constructed and binned for each morphometric index. Then, each patient’s PDF was projected onto the cohort PDF, ensuring that each patient’s PDF was on the same scale for computing distance. In the second case, the median of each computed index per patch and per WSI was first aggregated across the entire cohort to construct stable clusters. Once stable clusters (e.g., vocabularies) across the entire cohort were constructed, each WSI was represented in terms of the frequencies of each cluster.

#### 2.5.2. Distance Metrics and Clustering

Clustering based on the dictionary- and PDF-based methods are summarized below. This is an important step since using the first-order statistics (e.g., mean) of the PDF did not reveal any significant biomarkers.

The dictionary method (e.g., alphabet), which involves using the Euclidean distance to compute distances between computed features, is advantageous due to its simplicity. To cluster data, pairwise distances were computed and consensus clustering was performed by varying the number of clusters from two to four. This particular implementation of consensus clustering injected noise in each iteration, which helped reveal more stable clusters. The clustering results based on optimal transport were visualized using a similarity matrix, cumulative density function (CDF), and Silhouette plots (a visual measure of the quality of clusters), as shown in Figure 3. Pinhole images of each cluster for the eccentricity and cellularity index were shown in Figure 4 and Figure 5, respectively. For clustering, a random subset of 1000 samples was selected and k-means was iteratively performed on them. This process was repeated 10 times to obtain the centroids of the final clusters, which were determined by aggregating the median values corresponding to each sampled dataset for consensus clustering. The stability of each cluster was determined by the change in the CDF between the number of clusters and their silhouette scores. Subsequently, the learned alphabets were projected back into each WSI to create a patient signature based on the frequency of composition of each vocabulary, resulting in a vector (e.g., [0.34, 0.33, 0.33]) for three alphabets. This vector representation was then used as a continuous variable input to a Cox Hazard model to associate an increase in the percentage for each variable of the vector to a hazard ratio. Figure 6 illustrates this entire process.

Using the euclidean distance metric to measure the distance between the PDFs of two WSIs is inaccurate because it ignores the order of probabilities in a vector. In this study, we used the optimal transport method, also known as earth mover distance, to compute the distance between two PDFs. Optimal transport is a linear programming problem that we implemented using the Python Optimal Transport Toolbox. After computing pairwise distances between WSIs (i.e., PDFs), we performed linkage analysis [19,20] to reveal subpopulations. Figure 7 displays similarity matrices based on the optimal transport distance metric from PDF-based representations, and corresponding Kaplan–Meier curves (a probabilistic representation of a patient to survive up to a time) using three computed morphometric indices of nuclear size (e.g., area, left column) solidity (middle column), and total chromatin (right column).

#### 2.5.3. Statistical Analysis of Morphometric Indices for Biomarker Validation

Various statistical techniques, such as Kaplan–Meier curves, can be used to evaluate the predictive strength of each morphometric index on survival. However, the association between a computed index and survival may be biased if there are unaccounted variables (e.g., age) that strongly predict survival but are not balanced among the clusters formed by an index. To avoid such bias, we combined the age confounder with one morphometric index at a time in a regression model. For PDF-based representations of patients, we estimated a Kaplan–Meier curve for each index and then used the Cox regression model to estimate the hazard ratio and its *p*-value. However, when age was included, we observed evidence of a violation of the proportionality assumption in the Cox model (a statistical model for survival outcome with at least one predictor) [21], as hazards were not proportional with a *p*-value of 0.01. By including both age and age-squared, we found no evidence of a violation of the proportionality assumption, and an improved *p*-value. Therefore, we chose to use the likelihood ratio with age and age-squared in all our analyses.

The Likelihood-ratio test (LRT) is one of the three standard approaches for statistical hypothesis testing that evaluates the goodness of fit of two competing statistical models by comparing their likelihoods. In our study, we used the LRT to compare the Cox regression model, which included only age and age-squared (the “null” model), to the model that included age, age-squared, and the morphometric index (the “alternative” model). If adding an index, such as nuclear size, improved the model’s fitness compared to the model without the index, the *p*-value of the likelihood ratio test would be small (<0.05), indicating evidence that the index is significantly associated with survival even after controlling for age. Conversely, if the index did not improve the model’s fitness compared to using only age and age-squared, the likelihoods of the null and alternative models would be similar, and the likelihood ratio test would yield a large *p*-value (>0.05). When the LRT yields a small *p*-value, it provides evidence that the index is predictive of survival, but it does not provide information on the size of the effect or whether the range within the index is significantly different. Therefore, for each condition where the LRT *p*-value is less than 0.05, we also computed the 95% confidence intervals and tested pairwise differences between the hazard ratios corresponding to the levels of the variable of interest. In some cases, hazard ratios could not be estimated and were excluded from figures because either the patient’s survival time was close to zero or censored.

#### 2.5.4. Computing Resources

The machine learning models were trained on a local server, which was equipped with 8 NVIDIA GeForce RTX 2080 Ti GPUs, each with 12GB of RAM, 256 GB of RAM, and a 64-core CPU. The model development and validation were performed using python 3 and the TensorFlow 2.2 framework. The source code has been made available at https://github.com/gwinkelmaier/GBM-biomarkers (accessed date 31 March 2023).

## 3. Results

### 3.1. Biomarker Discovery

#### 3.1.1. Biomarkers of Nuclear Morphometric Indices

Table 1 display age-adjusted biomarkers based on PDF- and dictionary-based representations, respectively. The PDF representation includes biomarkers such as average chromatin content (e.g., HOD), nuclear size, solidity, and total chromatin (e.g., total HOD) per nucleus. Notably, our results show that the method in [17], from Navab’s Lab, yielded a statistically significant biomarker approximation of chromatin content, whereas the classical method based on known densities [18] did not. Additionally, the dictionary method uncovered eccentricity (e.g., elongation) and cellularity. Figure 8 illustrates the forest plots for predicted morphometric indices based on PDF-based methods without preconditioning. Finally, Table 2 displays predicted morphometric indices from both PDF and dictionary-based representations. This was achieved by (a) estimating the parameters of a Cox-Hazard model by integrating morphometric indices from both representations and (b) comparing the learned model with the baseline model of only age and age-squared, leading directly to a *p*-value.

#### 3.1.2. Biomarkers of Morphometric Indices Preconditioned on Genomics Signature

The same set of tests from the previous section was applied following stratification based on published genomics subtypes [2] or EGFR expression (e.g., high versus low expression rendered by linkage analysis).

There are four significant genomic subtypes: mesenchymal, proneural, neural, and classical. Table 3 display age-adjusted biomarkers based on these subtypes, computed from both PDF- and dictionary-based representations. The corresponding forest plot for the classical subtype in the PDF-based representation is shown in Figure 9, and Table 4 shows predicted biomarkers from the combined representation. For instance, nuclear size is a biomarker for the neural subtype, total HOD is a biomarker for the classical subtype, and solidity is a biomarker for the neural and mesenchymal subtypes. Therefore, there is evidence that each genomic subtype can highlight specific biomarkers, leading to further stratification of the patient population.

Aberrant overexpression of EGFR is a dominant feature of GBM. As a result, patient data were initially stratified based on low and high EGFR expression, followed by the proposed analysis as described in the Methods section. Table 5 and Table 6 present predicted biomarkers for the subpopulation of patients with high or low EGFR expression. Table 7 shows predicted biomarkers based on combined models of the PDF- and dictionary-based method preconditioned on EGFR expression. Note that Table 7 is quite similar to Table 2 as only a subset of patients with matched transcriptome data was used in this analysis. Additionally, Table 7 do not share a morphometric index, which serves as an internal control. The corresponding forest plot of morphometric biomakers, computed from the PDF representation and preconditioned on low EGFR expression, is shown in Figure 10.

## 4. Discussion

This manuscript presents our extended and applied methodologies for identifying biomarkers in GBM while taking into account tumor heterogeneity. Our approach involves using age-adjusted representations of nuclear morphometric features or their organization in WSIs and utilizing linear associations to improve interpretation while reducing the number of parameters. We suggest that incorporating the Cox Hazard Model in the loss function (as done in [22]) increases the likelihood of finding associations by noise or chance. Therefore, we advocate for using linear associations instead, as they offer simplicity and using a single computed index at a time improves interpretability and robustness. Lastly, we statistically explored viable patient pathology stratifications by preconditioning on either EGFR expression or genomic subtypes.

The methodological innovations used in this study included image normalization, representation and distance metrics, integration of age as a confounder, and the utility of age-squared to satisfy the proportionality assumption of statistical models. Image normalization is an important step in using HOD as a biomarker because of technical variations in sample preparation and staining. Navab’s Lab provided the foundation for normalizing each patch in a WSI to a reference template. NMF enabled the readout of the HOD per nucleus using the nuclear mask computed from the segmentation step. Nuclear masks provided the required initialization for NMF, making convergence rapid without needing a ranking based on the blue channel. Alternatively, image normalization based on classical color deconvolution [18] did not reveal HOD as a biomarker. Because tumors are heterogeneous, the study designed representations based on the PDF or dictionary-based method for each morphometric index in WSIs. The PDF method represents each WSI in terms of its own nuclear morphometric architecture. In contrast, the dictionary-based method represents each WSI in terms of learned alphabets that represent the entire cohort. The PDF method identifies biomarkers that are globally persistent within a WSI because attributes with a low frequency of occurrence can diffuse with the PDF representation. The two methods are complementary. While the distance measures for the dictionary-based method can be Euclidean, a distance measure based on optimal transport is introduced to compute distances between pairwise PDFs followed by linkage analysis. In GBM, age is the single most important predictor of the outcome. In the absence of the utility of age as a confounder, many biomarkers are either erroneously predicted or the proportionality assumption of the Cox Hazard model is violated. However, by using both age and age-squared, a more rigorous statistical analysis can be achieved. For example, since both the dictionary- and PDF-based methods predicted the age-adjusted HOD index as a biomarker, this index is more stable. A larger value of HOD content, a surrogate index for nuclear hyperchromasia, is consistently associated with a higher hazard ratio, which could be due to a higher rate of proliferation. Using the dictionary-based method, increased cellularity (e.g., hypercellularity) is also associated with a higher rate of proliferation and a higher HR. Another biomarker in the PDF method is solidity, which is a surrogate index for pleomorphism. Figure 8 suggests that lower pleomorphism corresponds to a better HR, where higher pleomorphism is associated with a higher tumor grade and worse prognosis. These observations are consistent with key diagnostic points in GBM, including cytological criteria of astrocytoma (e.g., GFAP, spindle-shape nuclei) and anaplasia (e.g., hypercellularity, pleomorphism, nuclear hyperchromasia) [23].

In conclusion, the computational pipeline has stratified tumor heterogeneity within the TCGA GBM cohort and has identified interpretable biomarkers that align with GBM diagnostic criteria [23]. This pipeline provides a valuable platform for evaluating emerging therapies for the treatment GBM patients. The same pipeline can also be applied to other tumor types. For example, it can be applied to WSI collected from Low-Grade Glioma (LGG) because of the similarities of the microanatomy. However, it may require additional extensions if it is to be applied to other organs, such as the breast or pancreas with glandular structures. In these types of organs, normal gland structures and stromal regions need to be delineated prior to the analysis of the tumor regions. This step requires another computational module that can be facilitated by annotation and training of the corresponding regions of the microanatomy.

## Figures and Tables

**Figure 1 cancers-15-02387-f001:**
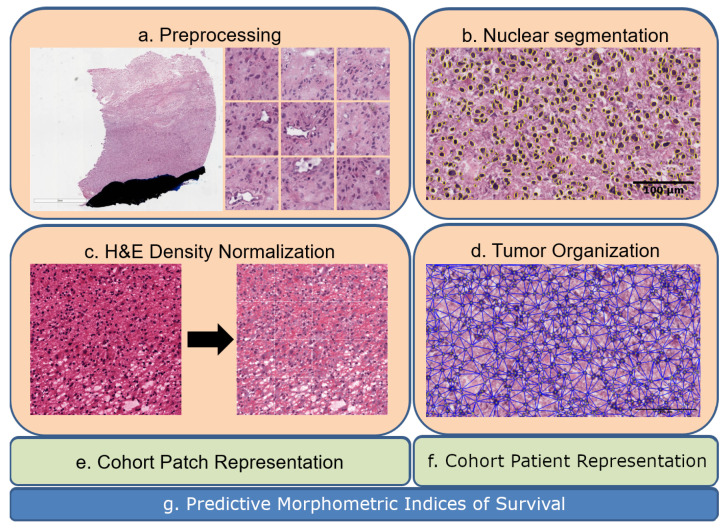
Eash WSI is represented in the context of tumor heterogeneity for biomarker discovery: (**a**) a WSI is partitioned to patches of 224-by-224, where each patch is analyzed for pen marks or other aberrations; (**b**) nuclei are segmented in patches; (**c**) H&E optical density is normalized in each patch; (**d**) nuclei organization is quantified in each patch; (**e**,**f**) computed indices from nuclei and their organizations are used for the dictionary- and PDF-based representations. (**g**) Predictive morphometric indices of survival are identified.

**Figure 2 cancers-15-02387-f002:**
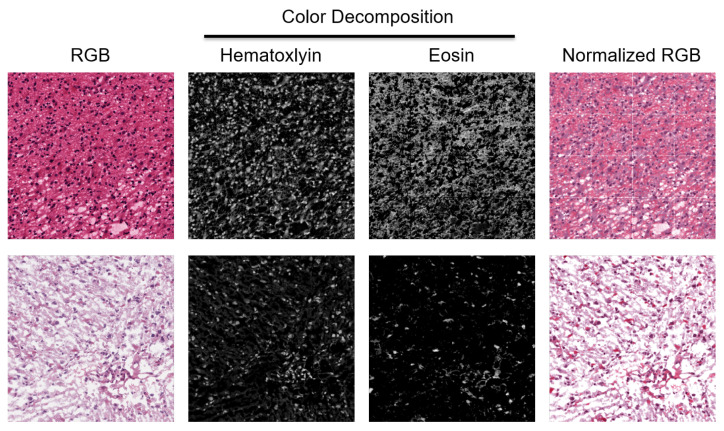
H&E stain is heterogeneous between patients. Two patches from two WSIs indicate a diverse staining signature. They are normalized for quantifying HOD and visualized in the RGB space.

**Figure 3 cancers-15-02387-f003:**
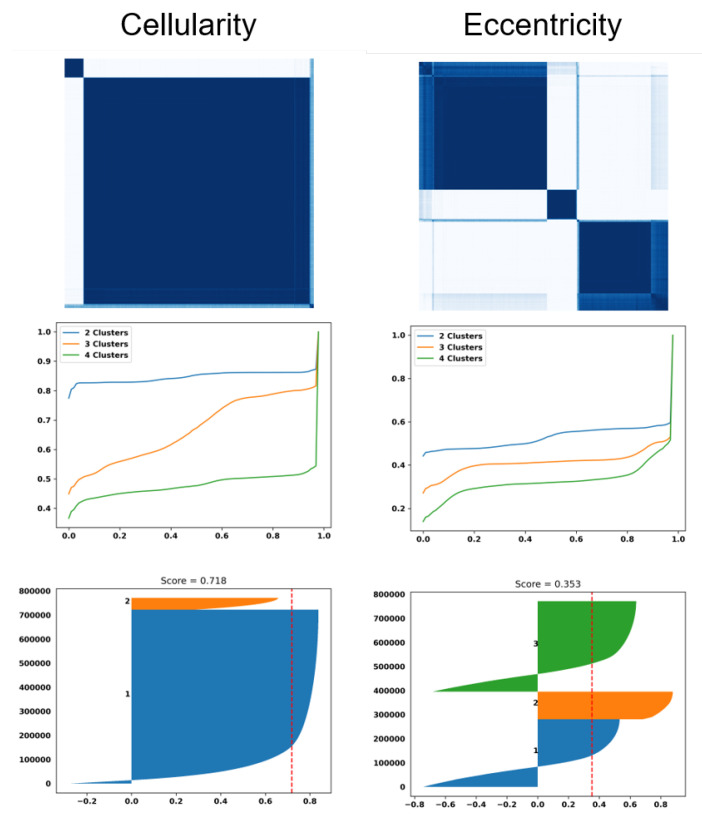
Dictionary-based learning identified two and three subpopulation (e.g., clusters) of patients based on cellularity and eccentricity indices, respectively. (top row): Computed similarity matrices; (middle row) the cumulative Density Function (CDF) of similarity matrices shows the quality of the number of clusters for each index (e.g., a flat horizontal line indicates a low number of misclassified samples between clusters). (bottom row) Silhouette plots of 800,000 randomly sampled nuclei show the similarity of patients within a cluster (e.g., a silhouette score less than 1) and a red dashed indicating the average silhouette score.

**Figure 4 cancers-15-02387-f004:**
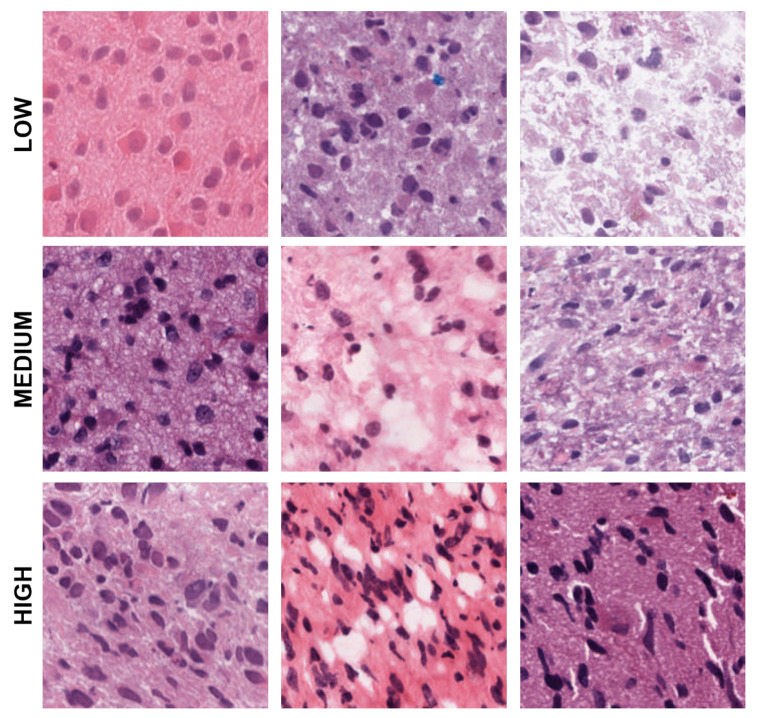
Representative patches showing low, medium, and high eccentricities corresponding to clusters 1, 2, and 3 from the dictionary-based method.

**Figure 5 cancers-15-02387-f005:**
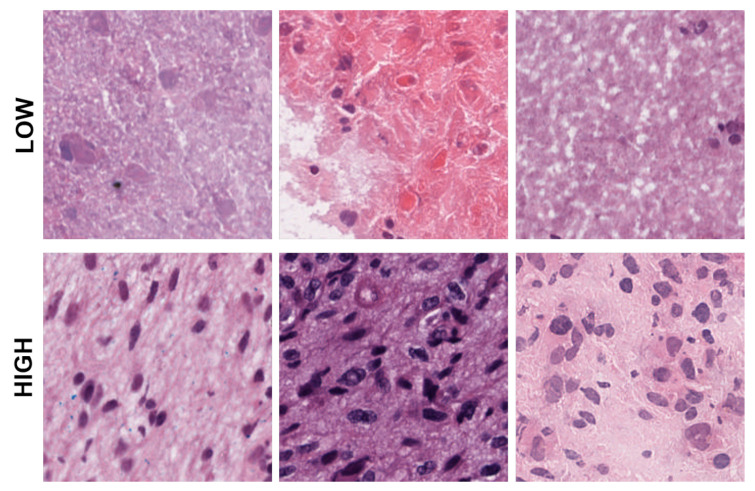
Representative patches showing low, and high cellularities corresponding to clusters 1 and 2 from the dictionary-method.

**Figure 6 cancers-15-02387-f006:**
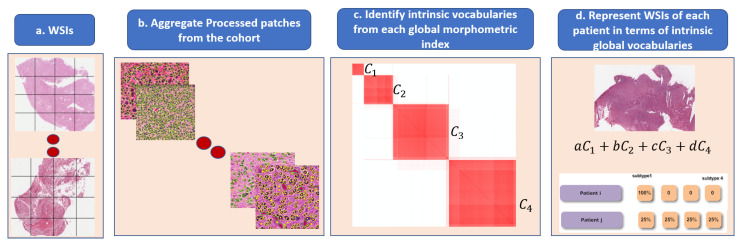
Steps in the dictionary-based method for representing heterogeneity: (**a**) each WSI is partitioned into patches; (**b**) each patch is quantified in terms of nuclear indices and organization; (**c**) each computed index (e.g., HOD content, nuclear size) is aggregated across the entire cohort for dictionary-based learning (e.g., alphabets, which are four in this example); and (**d**) each WSI is then represented as a composition of learned alphabets.

**Figure 7 cancers-15-02387-f007:**
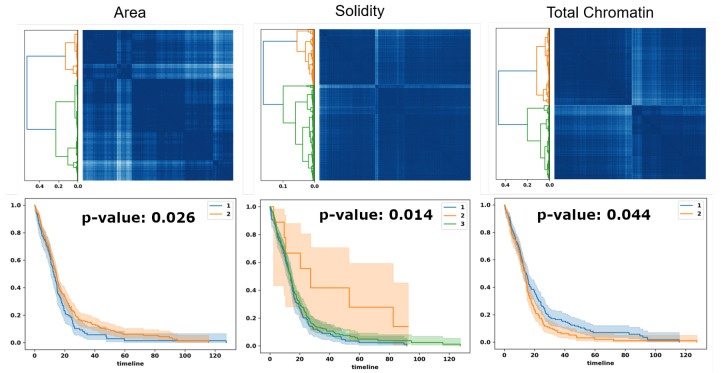
Optimal transport identifies subpopulations of patients, based on PDF representation, for survival analysis. Top row: similarity matrices identified by linkage analysis; Bottom row: Kaplan–Meier plots, hazard ratio, and computed *p*-values for three computed morphometric indices of nuclear size, solidity, and total chromatin.

**Figure 8 cancers-15-02387-f008:**
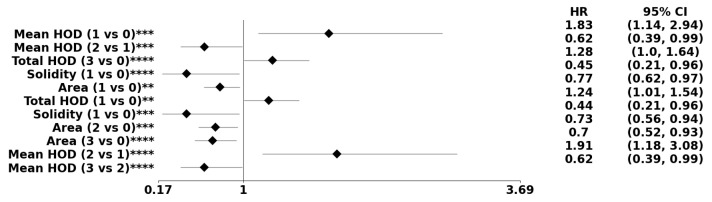
The forest plot indicates biomarkers associated with the subpopulation at risk using the PDF-based representation without any genomic preconditioning. The asterisks **, ***, and **** denote the number of stratifications per morphometric index.

**Figure 9 cancers-15-02387-f009:**
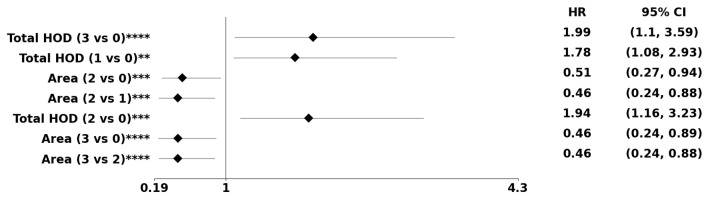
Using the PDF method, pre-conditioned on the classical subtype, the forest plot indicates the subpopulation at risk. The asterisks **, ***, and **** denote the number of stratifications per morphometric index.

**Figure 10 cancers-15-02387-f010:**
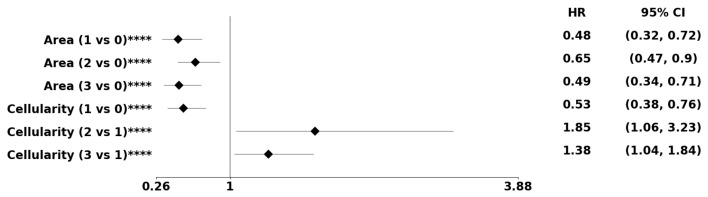
Using the PDF method, pre-conditioned on a high EGFR expression, the forest plot indicates the subpopulation at risk. For example, Area cluster two has an 52% decreased risk of death compared to Area cluster zero. The asterisks **** denote the number of stratifications per morphometric index.

**Table 1 cancers-15-02387-t001:** Predicted morphometric biomarkers and their *p*-values from patients in the TCGA-GBM cohort.

	Morphometric Index	Number of Clusters	*p*-Value
(**a**) PDF model			
	Area	2	0.026
	Area	3	0.016
	Area	4	0.013
	Mean HOD	3	0.016
	Mean HOD	4	0.006
	Solidity	3	0.014
	Solidity	4	0.007
	Total HOD	2	0.044
	Total HOD	3	0.037
	Total HOD	4	0.025
(**b**) Dictionary model			
	Cellularity	2	0.008
	Cellularity	3	0.040
	Eccentricity	2	0.002
	Eccentricity	3	0.005
	Eccentricity	4	0.011
	Mean HOD	2	0.019

**Table 2 cancers-15-02387-t002:** Predicted morphometric biomarkers and their *p*-values for the combined model without genomic preconditioning.

Nuclear Morphometric Index	Number of Clusters	*p*-Value
Cellularity	2	0.025
Eccentricity	2	0.007
Eccentricity	3	0.013
Eccentricity	4	0.019
Mean HOD	4	0.028

**Table 3 cancers-15-02387-t003:** Predicted morphometric biomarkers for the PDF- and dictionary-based models preconditioned on genomic subtypes.

Nuclear Morphometric Index	Number of Clusters	*p*-Value			
		Neural	Proneural	Mesenchymal	Classical
(**a**) PDF method					
Area	2	0.021	-	-	-
Area	3	0.020	-	-	0.009
Area	4	0.018	-	-	0.006
Mean HOD	4	0.024	-	-	-
Solidity	3	0.006	-	-	-
Solidity	4	<0.001	-	0.009	-
Total HOD	2	-	-	-	0.019
Total HOD	3	-	-	-	0.008
Total HOD	4	-	-	-	0.008
(**b**) Dictionary method					
Area	4	-	-	-	0.040
Total HOD	2	<0.001	-	-	-
Total HOD	3	0.008	-	-	-
Total HOD	4	0.003	-	-	-

**Table 4 cancers-15-02387-t004:** Predicted morphometric biomarkers and their *p*-values for the combined model preconditioned on genomic subtypes.

Nuclear Morphometric Index	Number of Clusters	*p*-Value			
		Neural	Proneural	Mesenchymal	Classical
Area	2	0.04	-	-	-
Area	4	-	-	-	0.043
Mean HOD	4	-	0.031	-	-
Solidity	3	0.010	-	-	-
Solidity	4	0.004	-	0.048	-
Total HOD	2	0.001	-	-	-
Total HOD	3	0.012	-	-	-
Total HOD	4	0.004	-	-	0.036

**Table 5 cancers-15-02387-t005:** Predicted morphometric biomarkers for the PDF- and dictionary-based models preconditioned on patients with high EGFR expression.

	Nuclear Morphometric Index	Number of Clusters	*p*-Value
(**a**) PDF model			
	Total HOD	4	0.048
(**b**) Dictionary model			
	Area	2	0.007
	Area	3	0.009
	Area	4	0.025

**Table 6 cancers-15-02387-t006:** Predicted morphometric biomarkers for the PDF- and dictionary-based models preconditioned on patients with low EGFR expression.

	Nuclear Morphometric Index	Number of Clusters	*p*-Value
(**a**) PDF model			
	Area	4	0.031
	Cellularity	4	0.018
(**b**) Dictionary model			
	Cellularity	2	0.035
	Total HOD	2	0.001
	Total HOD	3	0.003
	Total HOD	4	0.009

**Table 7 cancers-15-02387-t007:** Predicted morphometric biomarkers and their *p*-values for the combined model preconditioned on the EGFR transcript.

	Nuclear Morphometric Index	Number of Clusters	*p*-Value
(**a**) Biomarkers for patients with matched transcriptome data			
	Cellularity	2	0.047
	Cellularity	3	0.033
	Cellularity	4	0.019
	Eccentricity	2	0.040
	Mean HOD	3	0.010
	Mean HOD	4	0.004
(**b**) Biomarkers of patients stratified with high EGFR expression			
	Area	3	0.004
	Area	4	0.005
	Cellularity	3	0.031
	Cellularity	4	0.009
(**c**) Biomarkers of patients with low EGFR expression			
	Cellularity	2	0.034
	Cellularity	4	0.018
	Mean HOD	3	0.015
	Mean HOD	4	0.021
	Total HOD	2	0.001
	Total HOD	3	0.002

## Data Availability

https://github.com/gwinkelmaier/GBM-biomarkers (accessed on 31 March 2023) contains code and newly annotated images.

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
