# Peer review of "Biomarkers of Tumor Heterogeneity in Glioblastoma Multiforme Cohort of TCGA"

_cancers, 2023, doi:10.3390/cancers15082387_

Round 1
Reviewer 1 Report
Very original manuscript.
Recommended for publication.
1. What is the main question addressed by the research? Diagnostic and relevance for automatized verifications in clinics 2. Do you consider the topic original or relevant in the field, and if so, why? Application in clinics 3. What does it add to the subject area compared with other published material? New and original approach with quantification modelism 4. What specific improvements could the authors consider regarding the methodology? It is a new approach made for clinics 5. Are the conclusions consistent with the evidence and arguments presented and do they address the main question posed? The conclusions are relevant with all figures displayed ands explanations provided 6. Are the references appropriate? The limitation of references showed iy is a new approach in methodology for assisting in oncology diagnostics 7. Please include any additional comments on the tables and figures. I do not have any, unless another reviewer who is a neuropathologist would provide otherwise.
Author Response
None is requested
Reviewer 2 Report
In this article, the authors designed a way to identify biomarkers from the Glioblastoma Multiforme cohort of whole slide images from TCGA archive based on the previous method established by other group (Navab’s Lab). It's interesting study. However, I have a couple of concerns. First, can this way be generalized to analyze the WSI of all the cancer types rather than GBM only? Second, what kind of software or tools needed for this analysis? Third, there are a lot of professional terms, which were not defined clearly in the method section of the manuscript. It is hard for the readers to follow up.
Author Response
The authors appreciate thoughtful comments made by reviewers. Changes in the manuscripts are tracked in the resubmission.
- Can this be generalized to the analysis of all WSI or just GBM? We opted to focus on GBM because previous research did not render a biomarker for this tumor type. This is because “age” is the single most confounder in predicting survival in GBM. However, by rigorous normalization of pathology images and definition of metrics for heterogeneity, one can define statistically significant biomarkers for this tumor type. The same pipeline can also be applied to other tumor types, which may require further extension. To start, the same pipeline can be applied to WSI collected from Low Grade Glioma. Other tumor types, such as pancreatic or breast, require further extensions to exclude stroma regions. The conclusion has been updated to reflect this response.
- What kind of software tools are needed for this analysis? The software is mostly written in Python and Matlab, and they have been registered with Github for public distribution. The machine learning framework uses Tensorflow. The method section has been updated.
- There are undefined technical terms in the Method section. We have reviewed and revised each section to make the manuscript self-contained. We also improved on key concepts and readability.